# Development and Design of an Online Quality Inspection System for Electric Car Seats

**DOI:** 10.3390/s24217085

**Published:** 2024-11-03

**Authors:** Fangjie Wei, Dongqiang Wang, Xi Zhang

**Affiliations:** 1College of Mechanical and Electrical Engineering, Zhongyuan University of Technology, Zhengzhou 450007, China; 2022104098@zut.edu.cn; 2Advanced Textile Equipment Technology Cooperation Center, Zhongyuan University of Technology, Zhengzhou 450007, China; 3School of Mechanical Engineering, Zhengzhou University of Science and Technology, Zhengzhou 450007, China; 6541@zut.edu.cn

**Keywords:** electric car seats, LabVIEW, online inspection system, data acquisition, abnormal noise, electrical performance

## Abstract

As the market share of electric vehicles continues to rise, consumer demands for comfort within the vehicle interior have also increased. The noise generated by electric seats during operation has become one of the primary sources of in-cabin noise. However, the offline detection methods for electric seat noise severely limit production capacity. To address this issue, this paper presents an online quality inspection system for automotive electric seats, developed using LabVIEW. This system is capable of simultaneously detecting both the noise and electrical functions of electric seats, thereby resolving problems associated with multiple detection processes and low integration levels that affect production efficiency on the assembly line. The system employs NI boards (9250 + 9182) to collect noise data, while communication between LabVIEW and the Programmable Logic Controller (PLC) allows for programmed control of the seat motor to gather motor current. Additionally, a supervisory computer was developed to process the collected data, which includes generating frequency and time-domain graphs, conducting data analysis and evaluation, and performing database queries. By being co-located with the production line, the system features a highly integrated hardware and software design that facilitates the online synchronous detection of noise performance and electrical functions in automotive electric seats, effectively streamlining the detection process and enhancing overall integration. Practical verification results indicate that the system improves the production line cycle time by 34.84%, enabling rapid and accurate identification of non-conforming items in the seat motor, with a detection time of less than 86 s, thereby meeting the quality inspection needs for automotive electric seats.

## 1. Introduction

With the rapid development of the automotive industry, cars have become an indispensable part of people’s live [1]. As one of the most frequently interacted components in a vehicle, the design of car seats, particularly in terms of comfort and convenience, has become a leading trend in the industry [2,3,4]. Traditional fixed seats can no longer meet the increasing demands for comfort and convenience, making electric seats, which offer both, the mainstream choice [5]. These electric seats integrate multiple functions such as electric adjustment, ventilation, massage, and heating, significantly enhancing the user experience. However, as seat functionalities increase, so does the rate of potential malfunctions. Moreover, vibration and noise are common and unavoidable phenomena in all operating equipment, and car seats also generate varying degrees of vibration and noise during operation [6,7]. When the vibration frequency falls within the range of 20 Hz to 20 kHz, it produces audible noise that can distract drivers and passengers, affecting both physical and mental well-being, and potentially compromising driving safety and comfort [8,9,10]. To address these issues, seat manufacturers typically conduct quality inspections before the seats leave the factory [11]. The quality of a seat is assessed based on the noise generated during operation and the motor’s working condition [12,13], with the motor’s performance being determined by monitoring current levels [14].

To improve seat quality, Du [15] and Ramalingam [16] et al. analyzed the impact of the vibration characteristics of automotive seat systems on ride comfort but did not analyze the noise generated during seat operation. Kim, S. et al. used statistical methods to analyze the noise sources of electric seat sliders but did not assess the performance of abnormal noises [17]. Wu et al. conducted tests on the electrical functions of automotive power seats using PLC as the core controller. However, due to the limited number of seat functions, they did not test seat ventilation and massage functions, focusing solely on electrical functions without addressing abnormal noise performance [18]. Huang et al. performed fault diagnostics on the noise generated by three motors in six actions of automotive power seats, but the number of motors tested was limited. In contrast, this study tested 13 actions, including massage and ventilation, with support for further seat function expansion [19]. Su et al. used LabVIEW to detect and evaluate the noise of automotive seat motors, but the testing was performed in an inefficient offline mode, without evaluating the electrical functions of the seats [20]. Wu et al. proposed a denoising method based on singular value decomposition and stochastic resonance to detect seat noise, but the tests were not conducted in an anechoic chamber [21]. In summary, current automotive power-seat testing technologies typically separate electrical function testing from abnormal noise performance testing, and most tests are conducted offline. This results in more complex procedures on production lines, inefficient use of space, low integration, and reduced efficiency [22]. Additionally, many abnormal noise tests do not utilize high-standard anechoic chambers, which leads to interference in test results. As power-seat functions continue to expand, existing testing technologies are increasingly inadequate to meet the demands of new function testing.

Based on this, this paper presents the development of an online quality inspection system for automotive electric seats using LabVIEW. The system is capable of conducting online synchronous detection of both seat noise performance and electrical functions, including new features such as massage and ventilation [23,24]. The testing is conducted in a high-standard soundproof room where environmental noise levels do not exceed 25 dB. This room is equipped with a dedicated seat testing branch that is co-located with the main assembly line via a connecting line, enabling online inspection of the seats. The system is designed with highly integrated hardware and software components that interact through a PLC and the LabVIEW testing program [25]. It utilizes a sound level meter and current sensor to collect noise and current signals during seat operation. Additionally, the system performs real-time data processing and analysis, which includes waveform and numerical displays, data analysis and evaluation, and database queries [26,27]. Finally, the system was tested and validated in a real production environment.

## 2. Components of the Testing Platform

The testing platform for the online quality inspection system of electric car seats is shown in Figure 1. It mainly consists of a PLC, control cabinets, an anechoic chamber, the electric car seat under test, a mechanical fixture tray device, a subline, and an industrial computer. To improve testing efficiency and accommodate four workstations, the platform utilizes two control cabinets, each capable of controlling two workstations, allowing for the simultaneous operation of all four during testing. The control cabinets are equipped with various components, including a PLC module, safety relays, low-voltage electrical devices, an uninterruptible power supply (UPS), LED lighting, and cooling fans. To optimize layout and facilitate worker operation, the industrial computer is housed inside the control cabinet, while the display screen is located within the testing line inside the anechoic chamber. The display screen is connected to the industrial computer via an HDMI cable.

The subline for the online quality inspection of electric car seats is connected to the main production line via a transfer line. The seats are fully assembled through a series of processes on the main production line. Once assembly is complete, the seats enter the anechoic chamber testing line via the transfer line. After completing all the required tests, the seats are returned to the main production line through another transfer line, thus completing the online inspection process. Compared to offline testing, the advantage of integrating online inspection lies in its ability to provide real-time monitoring and feedback. This allows for the timely detection and correction of production issues, improving testing efficiency and integration while saving layout space. The connected system reduces manual intervention through efficient automation, continuously monitoring the entire production process to ensure that every product undergoes inspection. This minimizes the defect rate and need for rework, ultimately reducing costs. Centralized data management and analysis help optimize production processes, enhance overall production efficiency, and improve product consistency and reliability through standardized testing procedures.

During the seat testing process, seats can easily tip over when they reach extreme positions, which can affect testing accuracy and efficiency. Therefore, it is necessary to design a universal fixture tray that can secure the seat and be compatible with the main production line. The fixture is mounted on the subline and moves to the next workstation via a transition motorized roller on the line, ensuring the seat remains stable throughout the testing process.

The environmental noise in seat production workshops typically ranges between 70 dB and 95 dB, while the noise level during the operation of electric seats usually remains below 42 dB. To ensure accurate testing, it is essential to reduce external noise interference. Therefore, a high-standard anechoic chamber with environmental noise levels not exceeding 25 dB needs to be constructed for seat testing. Using SolidWorks 3D software (SolidWorks 2024 SP0.1), a 3D model of the anechoic chamber was designed, incorporating soundproofing materials and acoustic foam between the walls. The structure’s strength was simulated and analyzed to ensure stability and safety during actual use. Additionally, the ventilation system and lighting facilities were designed, integrating vents, air conditioning units, and lighting fixtures into the model to ensure that the ventilation and lighting met the required standards. Figure 2a shows the 3D model of the anechoic chamber (unit: millimeters), and Figure 2b shows a real-life image of the anechoic chamber, which accommodates four testing stations that support multiple testing modes, including full- cycle, half- cycle, and quarter-cycle pass-throughs.

The seats being tested are the driver and passenger seats of a particular electric vehicle model. These seats are controlled by seven motors, enabling 13 adjustment functions [28], as depicted in Figure 3 and Table 1, which show motor positions and operational conditions. During the electric seat testing process, sound level meters and current sensors are used to collect noise and current signals generated by seat movements. By analyzing the collected signals in both the time and frequency domains, the quality of the seats can be evaluated. To prevent signal frequency confusion during data collection, the sampling frequency must comply with Shannon’s theorem, which requires that $\omega_{s}\geq 2\omega_{M}$, where $\omega_{M}$ is the maximum frequency of the signal [29]. In this test, the sampling frequency is set between 20 Hz and 20 kHz, with $\omega_{s}$ significantly exceeding $2\omega_{M}$.

## 3. Detection System Composition and Principles

As shown in Figure 4, The hardware of the testing system primarily consists of a data acquisition card, display, PLC, sound level meter, current sensor, and industrial computer. The industrial computer serves as the core for data processing, while the display is used to show the testing interface and issue the start testing signal. The PLC is responsible for receiving the start signal, activating the motors and the LabVIEW program, monitoring the limit switch signals, and controlling the direction of the motors. The sound level meter and current sensor are responsible for transmitting the collected signals to the industrial computer. The testing system software mainly consists of LabVIEW software (NI LabVIEW 2023 Q3 (64-bit)), PLC drivers, and data acquisition card drivers. LabVIEW, as a software development platform, employs a graphical programming language and implements data acquisition, storage, processing, analysis, and display functions in a modular design. After installing the data acquisition card drivers, the LabVIEW environment automatically adds ADVANTECH DAQ functions, facilitating data acquisition and channel configuration.

The testing system workflow is illustrated in Figure 5, with the operating voltage set at 14 V ± 0.5 V. The testing items include sound pressure level (dB), loudness (sone), and current (A), with sone being derived from dB through an algorithm. All motors undergo two full-cycle tests, covering both forward and reverse rotations of the seat motors. The PLC communicates with the industrial computer via TCP. When the PLC receives the start testing command from the industrial computer, the seat motors begin to operate. The sound level meter and current sensor transmit the collected data directly to the data acquisition card, which performs A/D conversion and sends the data to the industrial computer. The LabVIEW testing program then processes the data in various ways, including waveform and numerical displays, data analysis and evaluation, as well as data storage and retrieval. In accordance with factory testing requirements, the processed data’s maximum values, averages, and thresholds are compared to assess the compliance of the electric seats.

Automating production lines in seat manufacturing is essential for boosting both operational efficiency and product quality. Serving as the cornerstone of the automation system, the PLC seamlessly integrates main and sub-assembly lines with its strong control functions and adaptable communication interfaces. Sub-assembly lines linked to the main line via the PLC streamline the production process through automation. The PLC consistently tracks the operational status of sub-assembly lines, synchronizing their activities with the main line to meet production demands. Optimizing PLC programming and system settings boosts production efficiency and reliability. As depicted in Figure 6, seats undergo a series of assembly processes on the main production line before reaching a junction where the interface line connects with the main line. A lifting mechanism raises the jig tray and secured seat, advancing them on transition rollers. Once the interface line retracts, the seat moves into the set testing position, with testing beginning as soon as the anechoic chamber’s pneumatic soundproof doors shut. After testing, the seat returns to the main line through a different interface line.

## 4. LabVIEW Program Design

The LabVIEW testing program consists of four main functional modules: system configuration, signal acquisition, data processing and analysis, and database querying. In the LabVIEW workflow, the system configuration module is first used to initialize and configure the acquisition devices. The signal acquisition module connects the sound level meter and current sensor to the corresponding input channels of the data acquisition card, ensuring compatibility between the sensor outputs and the acquisition card inputs, and enabling real-time collection of data such as sound pressure levels and current. The data collected by the sensors is transmitted directly to the acquisition card, where it undergoes A/D conversion before being sent to the industrial computer. The LabVIEW program processes and analyzes this data in real-time, generating graphical representations such as noise and loudness spectrums, one-third octave noise spectra, and time-varying motor current curves. The processed data, including the average and maximum values of noise, loudness, and current, along with waveform graphs, are then displayed in real-time on the front panel. The complete workflow of the LabVIEW detection program is illustrated in Figure 7.

### 4.1. LabVIEW and PLC Interaction

In conventional communication methods, developers must create specific programs to enable data transfer between PLCs and supervisory computers. The integration of Siemens S7-1200 series PLCs with LabVIEW greatly simplifies this process [30]. Using LabVIEW, PLCs can communicate with supervisory computers directly by reading and writing Data Block (DB) segments, eliminating the need for complex coding and Dynamic Link Libraries (DLLs), thus simplifying usage.

As depicted in Figure 8, the communication between LabVIEW and the PLC begins by establishing a link through the “S7input” module. Once connected, the program retrieves the target PLC memory address and specific data values from the “address” and “value” modules, respectively, the latter of which are sourced from an SQL database. Subsequently, the program writes these values to the designated PLC memory address using the “SiemensS7Net Write” module. Following the write operation, the program assesses its success through the “Operate Result” module, gathering potential error codes and detailed error information. If errors occur, the error handling mechanism addresses them, with outcomes relayed to the “error out” module. The program concludes its operation by terminating the PLC communication through the “S7output” module.

In practical applications, using LabVIEW to communicate with the S7-1200 simplifies data interchange and control command transmission between PLCs and supervisory computers. This method is straightforward, user-friendly, and offers excellent real-time performance. The combination of LabVIEW’s graphical programming and the Siemens S7 protocol enables more efficient development and implementation of industrial automation control systems. After scanning with a barcode reader, the PLC transmits seat identification information to the testing system. The system then receives current and voltage values from the data acquisition card and noise levels from the sound level meter. It also communicates with the PLC to control 13 different movements of the electric seat motors, including horizontal and vertical adjustments.

The scanning protocol, as shown in Figure 9, locks onto each seat using a unique identification code, allowing for traceability. It enables flexible extraction of barcode segments of varying lengths to identify the seat type, driver or passenger side, and other information. For example, “17, 0, 5” represents the configuration data for the 3C code, where the total length of the 3C code is 17 characters. The feature extraction starts at position 0 and captures the next 5 characters. The first two digits of this 5-character segment represent the seat type and whether it is the driver or passenger seat. Similarly, the configuration data for process codes and part codes also define the total length, starting position, and segment length for feature extraction. The advantage of this approach lies in its ability to accurately extract key information from barcodes without processing the entire code, thereby improving data handling efficiency and accuracy. By adjusting the extraction positions and lengths flexibly, the program can adapt to different barcode formats, reduce data redundancy, and optimize storage and transmission. After extracting these fields, the program formats the strings accordingly and calculates the feature positions and lengths based on the barcode rules. These feature positions and lengths are assigned to corresponding variables for subsequent data processing. Once scanning is complete, the front panel displays the test information for the car seat, including barcode data, product type, seat position, and configuration codes.

### 4.2. Signal Acquisition and Processing

In the testing system, analog signals are captured through current sensors and sound level meters and need to be converted into digital signals that the processor can handle. This conversion is typically done using a data acquisition card or an A/D converter. Common types of data acquisition cards include single-channel and multi-channel, as shown in Figure 10. Single-channel acquisition introduces a time delay between signals, reducing conversion efficiency. In contrast, multi-channel acquisition uses independent data acquisition cards, allowing simultaneous conversion, which improves efficiency and eliminates time delays. Therefore, multi-channel acquisition is used in the testing process to enhance performance.

#### 4.2.1. Noise Signal Acquisition Module

Each operation of the seat is motor-driven, generating various types of noise during operation [31]. Mechanical noise originates from friction, impacts, or imbalances among internal motor components, leading to irregular vibrations in the seat’s mechanical parts and housing, such as bearing noise and rotor imbalance noise [32]. Electromagnetic noise stems from the electromagnetic forces within the motor, causing radial vibrations in the stator core, particularly pronounced in motors with eccentricities or asymmetrical magnetic paths. Additionally, the typical structure of seat motors incorporates built-in dual fans, whose operation induces aerodynamic noise from air movement, impacts, and friction.

Currently, mainstream methods for noise detection include sound pressure level, sound intensity level, and vibration velocity. Typically, sound pressure level measurement is straightforward, yielding accurate and reliable results, and is directly performed using a sound level meter. Given its accuracy, this method meets the noise detection needs on production lines; hence, this paper employs sound pressure level to measure noise in automotive seats. The definition of sound pressure level $L_{p}$ is:(1)$$L_{p}=101g\left(\frac{p^{2}}{p_{0}^{2}}\right)=201g\left(\frac{p}{p_{0}}\right)dB$$

In the formula:

$L_{p}$—Sound pressure level (dB).

P—the effective value of sound pressure at a given frequency $P_{a}$.

$P_{0}$—Reference sound pressure value, which is $2\times 10^{-5}P_{a}$.

The sound level meter, composed of several components including a capacitive microphone, preamplifier, attenuator, amplifier, frequency weighting network, and RMS indicator, operates by converting sound into an electrical signal. This signal is then impedance-matched, amplified, and frequency-weighted to produce a numerical sound level value. Sound level meters are categorized into standard and precision types based on their performance; precision meters are noted for their higher sensitivity and stability, providing a more accurate reflection of sound characteristics in actual environments. For collecting motor noise signals, a BSWA MPA261600046 precision sound level meter was selected as the data acquisition module. It is positioned in alignment with the car’s direction, vertically above the seat line at a distance of 0.7 m from the seat cushion, as shown in Figure 11.

#### 4.2.2. Noise Signal Acquisition and Processing

Figure 12 illustrates the noise signal acquisition program, which begins by reading configuration information, including the data acquisition card address cDAQ1-SV1/ai0 and other relevant parameters. The program automatically retrieves this information through the configuration file path and reading functions, applying it to subsequent acquisition settings. The DAQmx Create Channel function is used to establish a data acquisition channel configured for acoustic measurement in Pascals, indicating that the signal being collected is sound pressure, with the channel set to cDAQ1-SV1/ai0. The program sets the sampling rate to 51,200 Hz, equating to 51,200 samples per second, and configures the sampling mode to “Continuous Samples” to ensure data continuity and real-time capture. A 0.1-s sampling window is defined, during which data is collected for 0.1 s in each sample. The sampling interval is set at 0.05 s, indicating a wait time of 0.05 s between each sample. The sample clock is configured using the Sample Clock, with the Onboard Clock selected and a sampling rate of 51,200 Hz to ensure precise sampling intervals. The buffer size is set to 10, allowing it to hold 10 samples to ensure data stability and continuity during acquisition. During data reading and processing, the program enters a While loop to continuously read the acquired data. The sound pressure signals are retrieved from the data acquisition channel using the DAQmx Read function and displayed in real-time on a waveform chart for easy monitoring. Data can also be further processed or stored for subsequent analysis.

As shown in Figure 13, the program initiates processing of the captured noise data by employing LabVIEW’s built-in FIR filter function to process the incoming signals. The filtering aims to eliminate noise at specific frequencies, which is crucial for enhancing signal quality and accuracy. Considering potential disturbances at the start and end of motor testing, the signal is trimmed before processing to analyze only the required segments. The program calculates the mean and root mean square (RMS) values of the trimmed signal, which are then converted to decibels (dB) to more intuitively display signal intensity. Subsequently, the program conducts a detailed analysis of the signal in both the time and frequency domains, extracting various spectral data including one-third octave spectra, octave spectra, power spectra, and power spectral density [11,33]. These analyses reveal the frequency characteristics and energy distribution of the signal, providing a basis for further noise control and analysis.

Ultimately, the processed data is visually displayed on the front panel using LabVIEW’s Waveform Chart and Graph controls. The display includes key information such as the maximum and average values of the processed signals and their waveforms, enabling users to visually observe changes and characteristics of the signals. These steps form a comprehensive noise data processing workflow, providing thorough support for noise analysis.

While decibels precisely measure the pressure level of sound, they do not accurately reflect the subjective perception of the human ear. The human ear perceives sound intensity in a nonlinear manner. The loudness measure (sones) better reflects this nonlinearity, aligning more closely with human auditory experience and providing a more accurate representation of how loud sounds are perceived. Therefore, converting decibels to sones enhances the accuracy of sound level assessments.

The formula to derive loudness levels from sound pressure levels is as follows:(2)$$L_{N}=40\times \log B_{f}+94$$

The loudness level$\;L_{N}$ (in sones) of a pure-tone with frequency f and sound pressure level $L_{p}$. The conversion relationship between loudness (sones) and loudness level is as follows:(3)$$L_{N}=40+33\ln N$$

Or
(4)$$N=2^{\frac{L_{N}-40}{10}}$$

In the formula: N is the loudness, $L_{N}\;$is the loudness level.
(5)$$B_{f}={\left[0.4\times 10^{\left(\frac{L_{P}+L_{U}}{10}-9\right)}\right]}^{a_{f}}-{\left[0.4\times 10^{\left(\frac{T_{f}+L_{U}}{10}-9\right)}\right]}^{a_{f}}+0.005135$$

$T_{f}$: Hearing threshold, measured in decibels (dB)

$a_{f}$: Loudness perception exponent

$L_{U}$: Amplitude of the normalized linear transfer function for 1000 Hz, measured in decibels (dB)

When reworking a defective electric seat, anomalies in the noise and loudness of the horizontal motor were identified through the retrieval of historical test data. The waveform, as shown in Figure 14a, displayed several peaks and troughs in the decibel signal amplitude between 45 and 50 dB, with a peak at 48 dB exceeding the system’s threshold of 42 dB. The fluctuations in loudness were similar to those in the decibel graph, as shown in Figure 14b, indicating a correlation between sound pressure level and loudness. The loudness amplitude also showed multiple peaks and troughs, ranging from 3 to 10, with a peak at 7.5, which exceeded the threshold, leading to a determination of non-compliance. Such issues could arise from rotor imbalance, uneven electromagnetic forces, or unstable motor mounting positions [34]. After re-securing the motor’s position and retesting, both noise and loudness values were below the threshold, resulting in a compliant outcome, as depicted in Figure 14c,d.

#### 4.2.3. Current Signal Acquisition Module

During testing, the data acquisition card cannot directly capture current signals, so a current sensor is required to convert these current signals into voltage signals that the acquisition card can process. Currently, there are two types of current sensors: traditional and through-hole. The traditional type involves a more complex installation and connection process. In contrast, the through-hole sensor offers numerous advantages, including high precision, low power consumption, a compact structure, and ease of installation. Therefore, the HDC-AA 22mm through-hole current sensor is selected for the testing process.

#### 4.2.4. Current Signal Acquisition and Processing

In the current signal acquisition program workflow, as depicted in Figure 15, the program begins by loading configuration files and setting the physical channel to “Dev1/ai0” via the “AI Address” module. Within the “Config” module, the AI voltage and internal clock parameters are configured, while the “Sample Clock” module locks the sampling rate at 1000 Hz using an onboard clock. The sampling mode is set to continuous, ensured by the “Continuous Samples” setting, with a buffer size of 10,000 samples configured in the “DAQmx Buffer” module to prevent data loss during processing. Once the setup is complete, the “DAQmx Start Task” function activates the data acquisition task. The “Error Out” module captures any operational anomalies, and the “No Error” module ensures the process runs correctly. This setup not only enhances the precision of data collection but also improves system stability and real-time performance through efficient buffering and error management mechanisms.

As shown in Figure 16, the current signal processing program focuses on noise reduction and offset correction to optimize signal quality. Initially, current waveform data is input through the “waveform” module and converted into an array format within a For loop to facilitate subsequent operations. A median filter is applied to suppress noise, effectively removing outliers while preserving the main trend of the data. This step is crucial for ensuring stable signal processing. The filtered data is displayed in real time on the waveform chart module, allowing users to monitor the effects immediately and verify the filter’s performance. Subsequently, the data is passed to a statistics module to compute the mean and standard deviation, providing basic statistical information about the signal. All processed data is stored in an array and output through an array output module at the end of the loop, facilitating further analysis or archiving. The entire program is designed to enhance the precision and utility of data processing, enabling effective monitoring and adjustment by the user through real-time data display and instantaneous statistical analysis, thereby achieving optimal data analysis results.

During the rework of a defective electric seat, an analysis of historical testing data pinpointed an abnormality in the backrest motor’s current. To focus on critical current changes, simplify data processing, and effectively isolate and diagnose potential issues, a 5-s segment was extracted from a longer waveform for analysis, as shown in Figure 17a. The current initially surged to a peak of 6.32, a rapid rise due to the motor’s instantaneous demand for current to generate the necessary torque upon startup. Subsequent fluctuations between 1.5 and 6 were severe and frequent, possibly caused by bearing damage, rotor bar cracks, or mechanical failures in the stator or rotor. After replacing the motor bearings and retesting the seat, the current returned to normal levels [35]. The waveform, illustrated in Figure 17b, showed a smaller initial peak and a subsequent pattern of relatively stable, lower-amplitude fluctuations compared to the abnormal motor waveform.

### 4.3. Information Visualization

To provide users with an intuitive way to monitor and analyze the current and noise data for multiple testing items, the program features a user interface, as shown in Figure 18. The interface consists of five main sections: the title bar, status bar, test list, operation prompts, and navigation buttons. The title bar includes information such as time, user, and access rights. The status bar displays the PLC communication status and power supply status. The test list contains the test items, values, and results. The operation prompts section provides barcode descriptions, operation instructions, current test data, and illustrations. The navigation buttons allow users to switch between the configuration interface, manual interface, and query interface.

When performing the testing operation, the user opens the program and clicks the “Start” button, which brings up the barcode scanning interface. The seat barcode is then scanned using a barcode scanner. Once the scan is completed, the test begins, sequentially evaluating the 13 motor operations. After each operation, the average and maximum values of current, noise, and loudness, along with waveform graphs and one-third octave spectra, are displayed in real time on the user interface. At the end of each operation, the system determines the result based on preset thresholds, as shown in Table 2. If the result is within the threshold, it is displayed as “Pass” (OK); if it exceeds the threshold, it is marked as “Fail” (NG).

### 4.4. Database Creation and Querying

The database creation and querying system is designed to store and manage the extensive data generated during the testing of automotive electric seats, ensuring data integrity and consistency. By establishing a database, the collected current and noise data, along with other relevant information, can be systematically stored, making data access and retrieval efficient and convenient. The database offers robust querying capabilities, allowing users to filter and analyze data based on various criteria, generate reports, and track historical records, thus aiding in the optimization of the testing system’s performance and decision-making processes. The design principles and workflow of the database creation and querying system are illustrated in Figure 19.

The overall database design plan is illustrated in Figure 20. The process begins with seat assembly, followed by the collection of noise and current data, which is then analyzed to determine whether the data is normal. If the data is normal, it is stored in the database, and corresponding database queries can be performed. If the data is abnormal, the seat undergoes a retest. If the retest still produces abnormal data, the production line is forced to stop, and the issue is tracked down. The abnormality in noise, current, or assembly is analyzed, followed by re-testing and re-assembly. This process ensures the quality of seat assembly and allows for timely correction and analysis when issues arise.

After generating Excel spreadsheets from the test results, the LabVIEW program interacts with the SQL Server database to create and manage database tables for effective storage and processing of the test data. As illustrated in Figure 21, the program uses two main SQL statements to create two data tables, each designed to meet specific data management requirements.

During the creation of an SQL table named “Master Table”, the overall design optimizes data management and efficiency through carefully selected fields and storage strategies. The “CREATE TABLE” command is used to define the table structure under SQL Server’s default schema “dbo”, ensuring standardization and consistency. The primary key, “id”, is an auto-incrementing integer that starts at 1 and does not allow null values, automatically assigning a unique identifier to each record. This simplifies data operations and maintenance, preventing duplication and conflicts. The “Barcode” field provides flexibility by allowing the storage of various barcode information to accommodate different data entry needs. The datetime fields, “time” and “endtime”, record the start and end times of the tests. The “type” field records the type of event or test, while the integer field “final_rslt” stores the final result of the test or event. The entire table is stored on the “PRIMARY” filegroup in the database, optimizing data access performance and ensuring data integrity. In the creation of the “Test Results” SQL table, the “Master Table_id” field is an integer, ensuring that each record can be accurately linked to the master table, maintaining data consistency and integrity. The “run_condition” field, defined as a variable character type, provides flexibility for recording test conditions or statuses, which aids in later data analysis and troubleshooting. The “rslt” and “value” fields store the numeric test result and its detailed description, making the data easy to query and analyze. The “lower_limit” and “upper_limit” fields store the range of test result values, which are crucial for setting and validating product quality standards, ensuring that product performance falls within the defined parameters.

Overall, this design not only automates and standardizes the data collection and storage process but also enhances the system’s data processing capabilities and reliability through a flexible database table structure. It ensures that the management and analysis of test data are efficient and accurate. The database supports multiple query methods, including searching by item, barcode, part number, seat position, assembly code, model code, configuration code, and test time. These query options allow users to quickly retrieve specific information from a large dataset, facilitating data analysis and decision-making. Additionally, the system supports data management and maintenance, enabling users to locate and delete duplicate records, update incorrect information, and ensure the consistency and integrity of the data.

## 5. Testing System Verification

As illustrated in Figure 22, the testing system has been validated in actual production. After assembly on the main production line, seats move to a designated position within an anechoic chamber, where operators initiate the testing sequence. The system adjusts 13 different actions of the electric seats via PLC communication, capturing real-time data on currents and noise during operation. Results are immediately displayed on monitors. Once testing concludes, seats that meet quality standards proceed to the next stage of production, while those that fail undergo retesting through a non-conformance process (NG) to determine their final quality.

The original production line employed separate testing methods for noise performance and electrical functionality and operated in an offline mode, with a production cycle of 31 Jobs Per Hour (JPH), where the pace directly impacted the overall efficiency and functionality of the production. To enhance efficiency and output, the expected rate was targeted to increase to 40 JPH, with each seat being tested in under 90 s. To ensure the statistical significance of the results, data from the production line’s testing system was continuously collected over 10 days, resulting in a current rate of 41.8 JPH.

Percentage Increase in Rate =$\;\left(\frac{New\;Rate-Original\;Rate}{Original\;Rate}\right)\times 100\%=34.84\%$.

Average testing time per seat at the original rate = 3600/Original Production Rate JPH ≈116 s.

Average testing time per seat at the current rate = 3600/Current Production Rate JPH ≈86 s.

With a rate of 41.8 JPH, which exceeds the target of 40 JPH, and an average testing time of 86 s, which is below the 90 s threshold, the integration of the online testing system has surpassed expectations. The rate has increased by 34.84%, significantly enhancing production efficiency.

## 6. Conclusions and Outlook

This paper presents the development of an online quality inspection system for automotive electric seats based on LabVIEW, effectively addressing issues such as the numerous testing procedures and low integration levels that impact the production cycle of seat manufacturing. Through a highly integrated hardware and software design, the system enables the synchronous online detection of both noise performance and electrical functionality in electric seats. Testing is carried out in a high-standard soundproof room, where ambient noise does not exceed 25 dB, effectively eliminating external noise interference. Online seat testing is achieved via a built-in detection branch within the soundproof room. The system was tested and validated in a real production environment, demonstrating that it can complete the inspection of a seat and provide results within 86 s. Compared to the previous method of separately conducting offline tests for noise performance and electrical functionality, the system increased production cycle efficiency by 34.84%, significantly simplifying the seat production line’s testing process, optimizing layout space, and enhancing both integration and testing efficiency. Additionally, the system allows for the timely reworking of defective seats, effectively preventing faulty products from entering the market, reducing waste and rework rates, cutting costs, and providing valuable technical support for improving the production quality of automotive electric seats.

With the rapid development of the automotive industry, this system has broad application prospects. Future research could include the detection of additional key parameters, such as temperature and pressure, to further enhance the system’s testing capabilities. Additionally, the data analysis algorithms within the LabVIEW system could be further optimized to improve data processing speed and accuracy. More field application tests could also be conducted to collect data from different production lines and environments, verifying the system’s stability and reliability to meet the ever-changing market demands. Furthermore, exploring new technologies, such as machine learning or artificial intelligence (AI), could further improve the quality and performance of automotive seats, contributing to the sustainable development of the automotive industry.

## Figures and Tables

**Figure 1 sensors-24-07085-f001:**
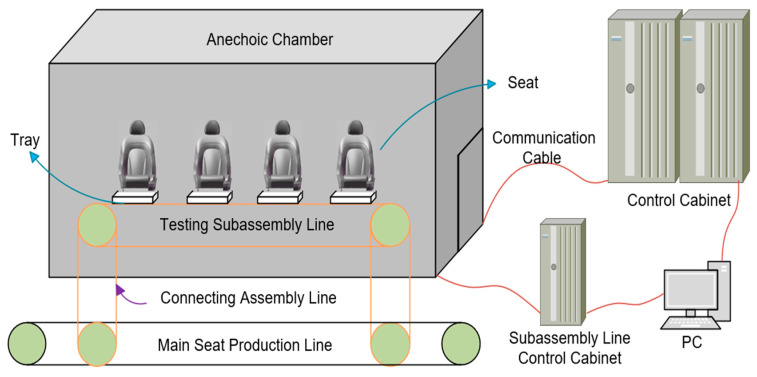
Structure diagram of the testing platform.

**Figure 2 sensors-24-07085-f002:**
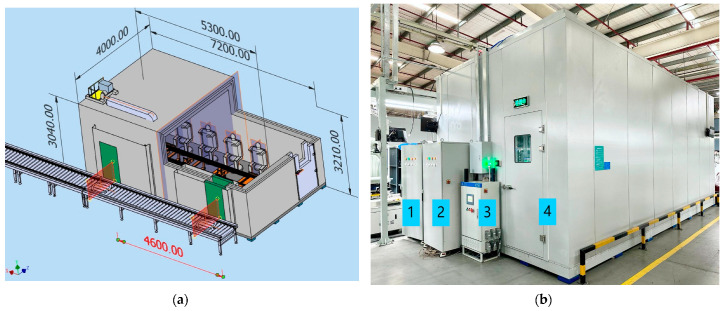
Anechoic chamber. (**a**) Anechoic chamber 3D model; (**b**) Anechoic chamber real-life image. Labels: 1—Control cabinet 1, 2—Control cabinet 2, 3—Subassembly line control cabinet, 4—Anechoic chamber.

**Figure 3 sensors-24-07085-f003:**
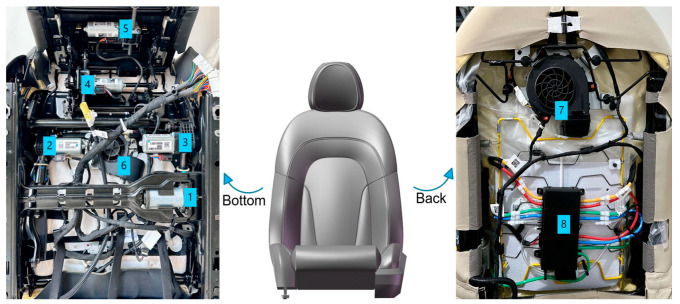
Seat motor positions. Labels: 1—Horizontal motor, 2—Backrest motor, 3—Front elevation motor, 4—Rear elevation motor, 5—Thigh support motor, 6—Seat cushion ventilation, 7—Seat backrest ventilation, 8—Lumbar support massage.

**Figure 4 sensors-24-07085-f004:**
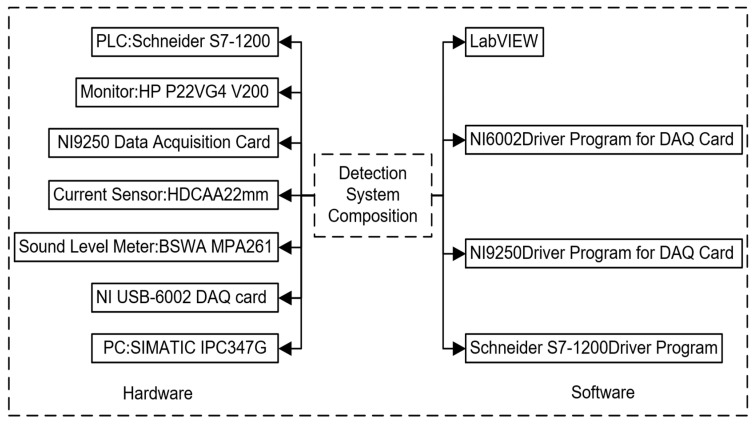
Software and hardware composition.

**Figure 5 sensors-24-07085-f005:**
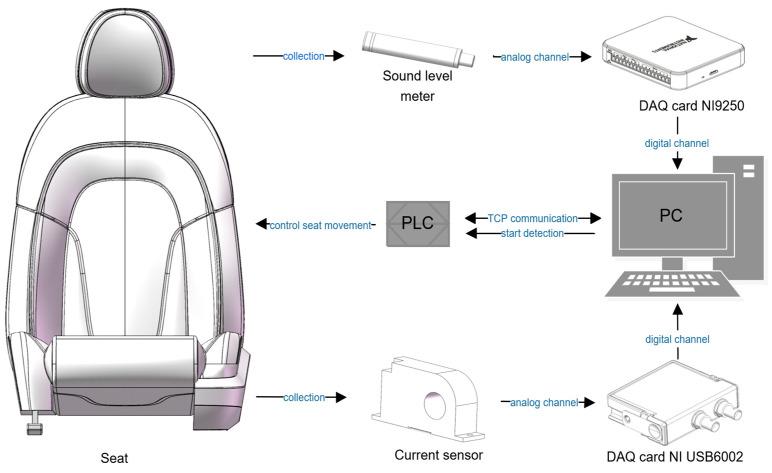
Detection system process.

**Figure 6 sensors-24-07085-f006:**
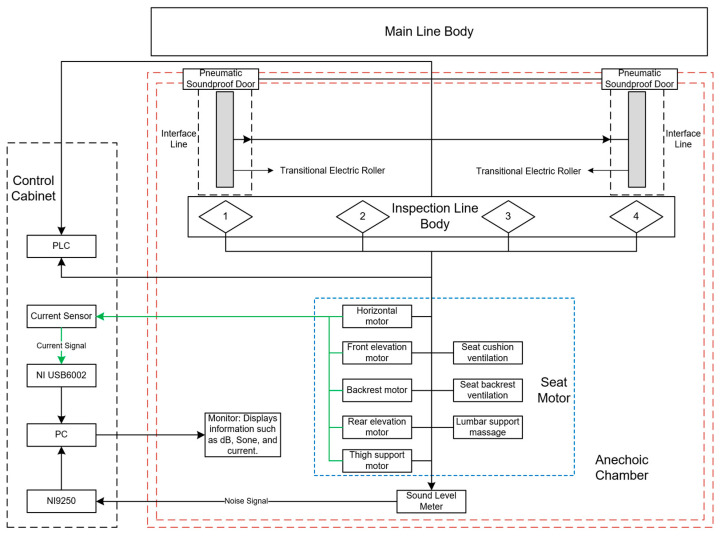
Testing system integration process.

**Figure 7 sensors-24-07085-f007:**
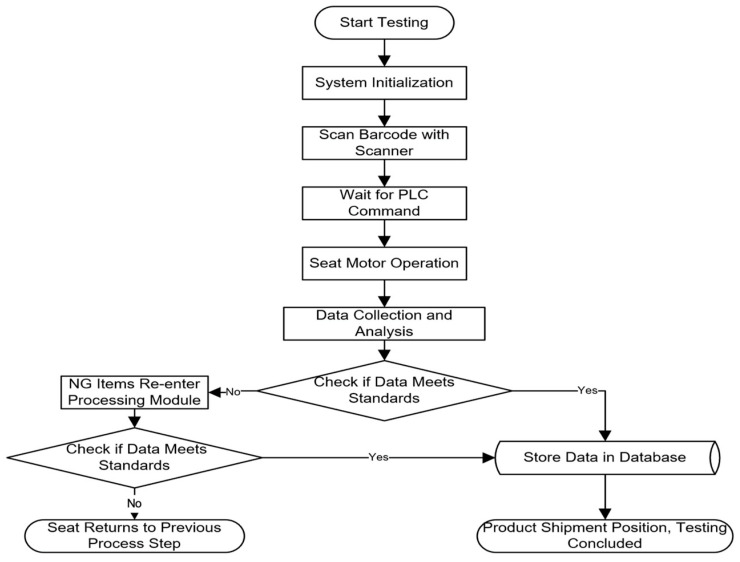
LabVIEW Software Operating Procedure.

**Figure 8 sensors-24-07085-f008:**
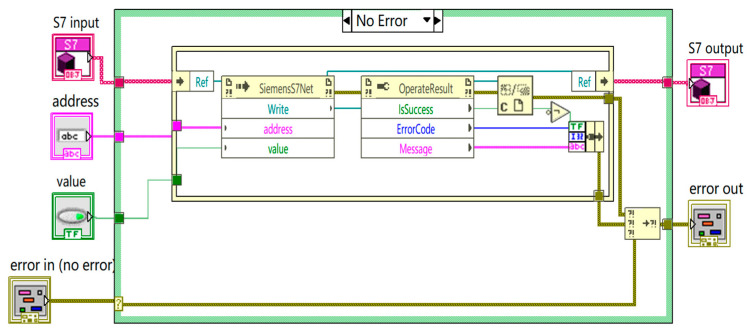
LabVIEW and PLC communication program.

**Figure 9 sensors-24-07085-f009:**
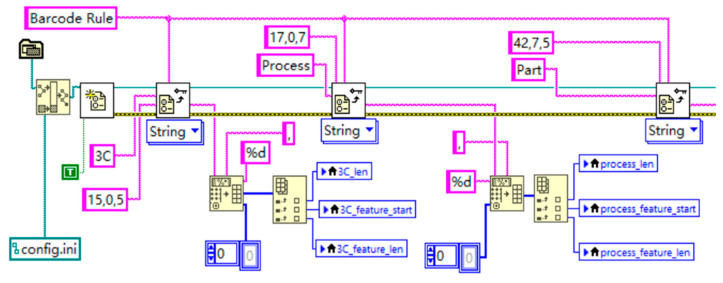
Barcode scanning protocol program.

**Figure 10 sensors-24-07085-f010:**
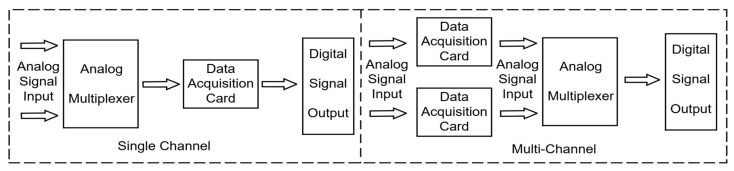
Single and multi-channel acquisition.

**Figure 11 sensors-24-07085-f011:**
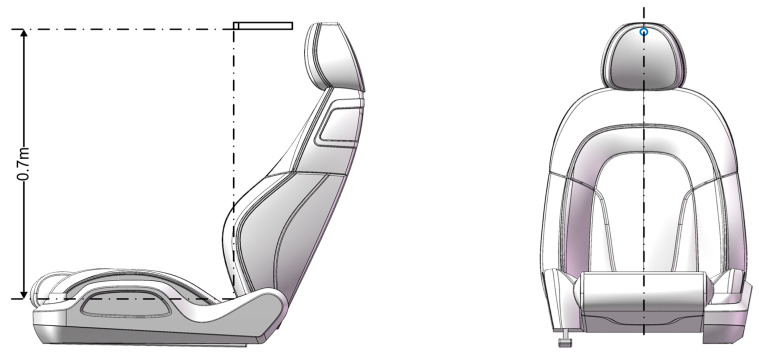
Sound level meter position.

**Figure 12 sensors-24-07085-f012:**
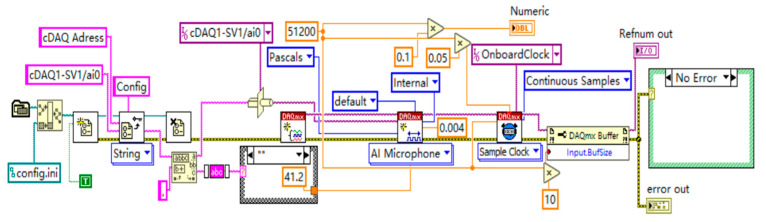
Noise signal acquisition program.

**Figure 13 sensors-24-07085-f013:**
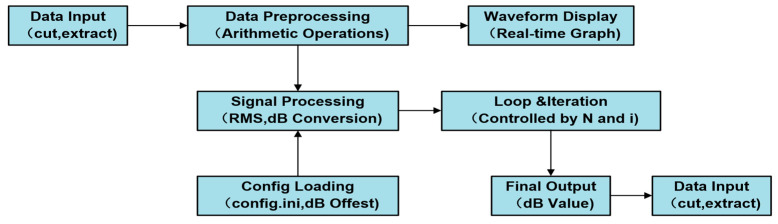
Noise data processing flow.

**Figure 14 sensors-24-07085-f014:**
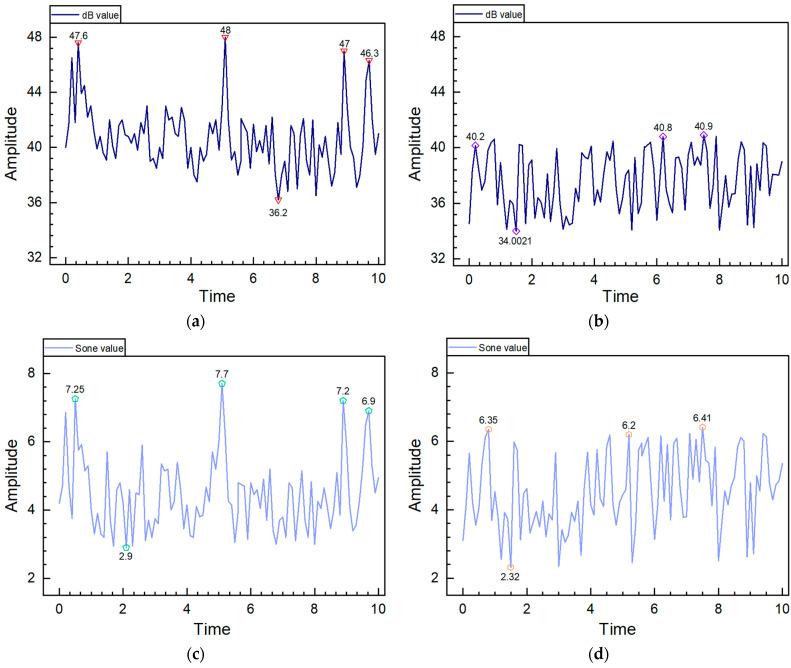
DB and Sone waveform charts. Labels: (**a**) Abnormal dB waveform; (**b**) Normal dB waveform; (**c**) Abnormal Sone waveform; (**d**) Normal Sone waveform.

**Figure 15 sensors-24-07085-f015:**
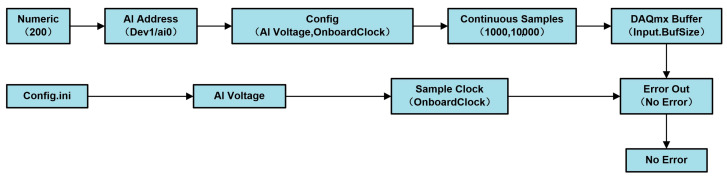
Current signal acquisition program workflow.

**Figure 16 sensors-24-07085-f016:**
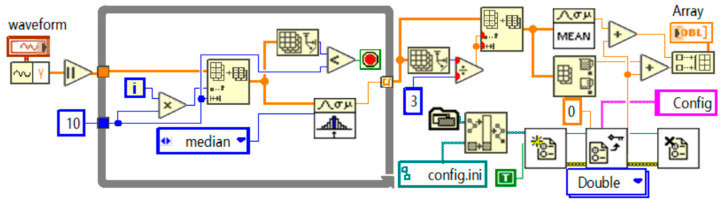
Current signal processing program.

**Figure 17 sensors-24-07085-f017:**
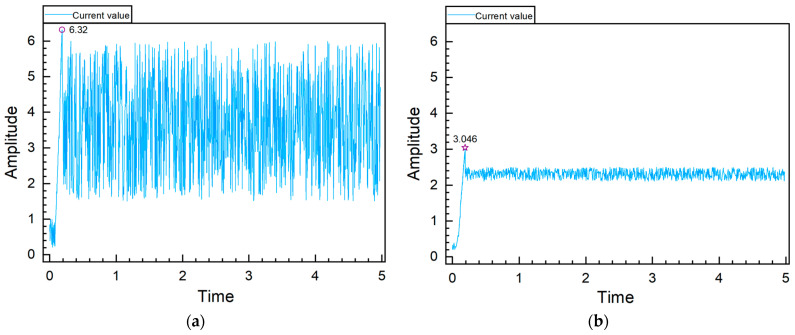
Current Waveform Diagram Labels: (**a**) Abnormal Current Waveform; (**b**) Normal Current Waveform.

**Figure 18 sensors-24-07085-f018:**
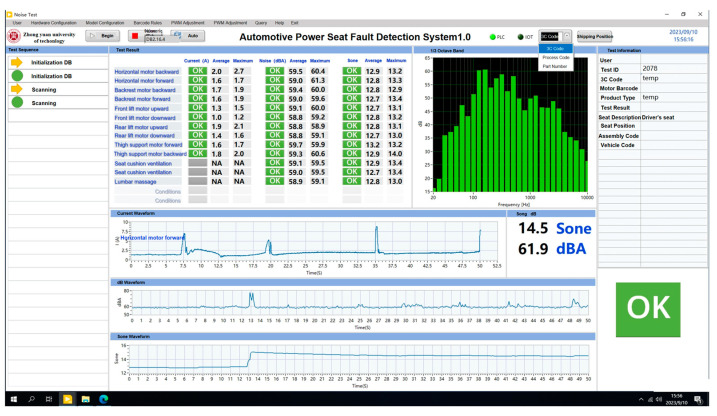
Main interface of the testing program.

**Figure 19 sensors-24-07085-f019:**
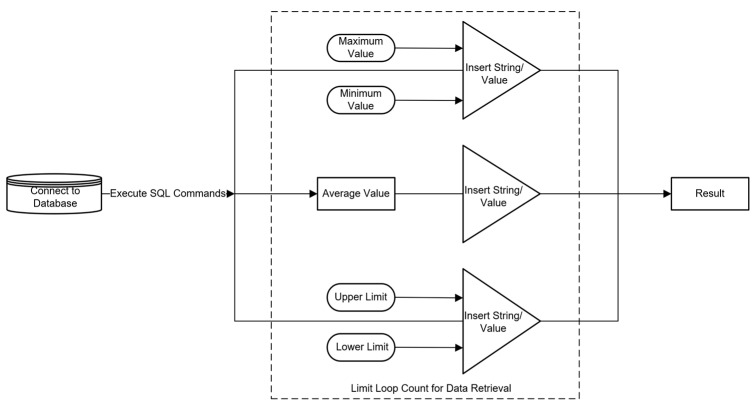
Database creation and query system design principle workflow.

**Figure 20 sensors-24-07085-f020:**
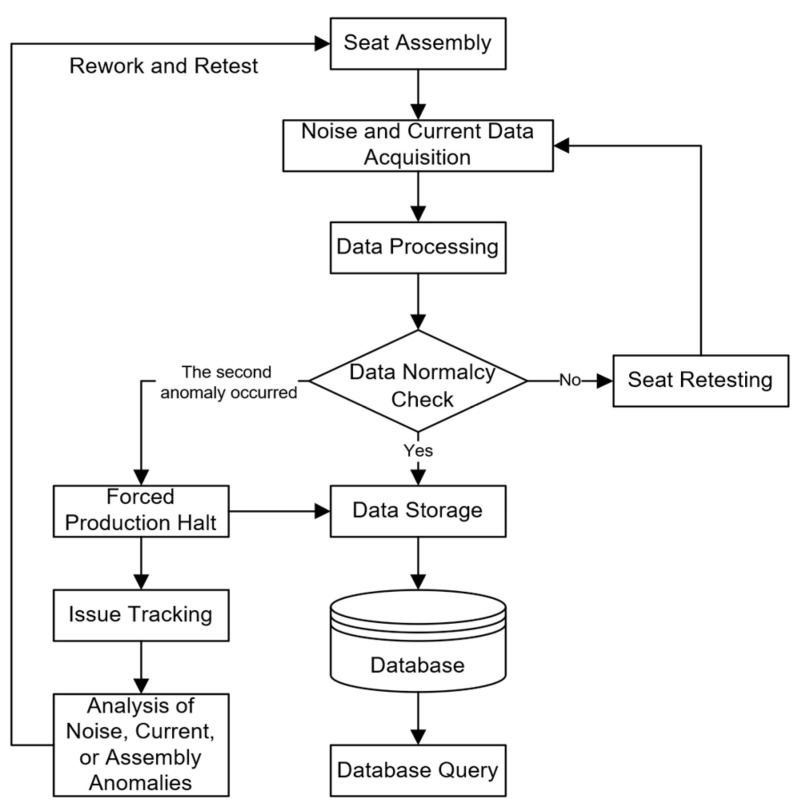
Overall database design plan.

**Figure 21 sensors-24-07085-f021:**
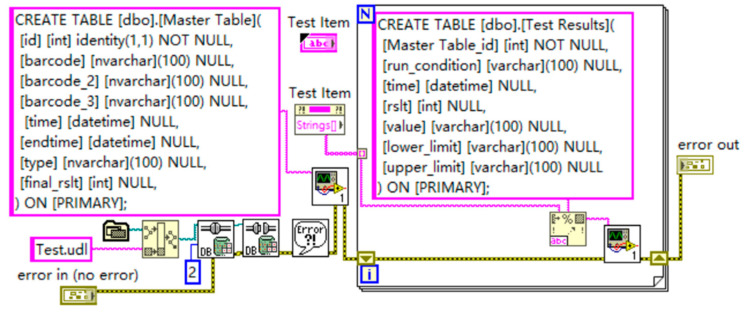
Database setup program.

**Figure 22 sensors-24-07085-f022:**
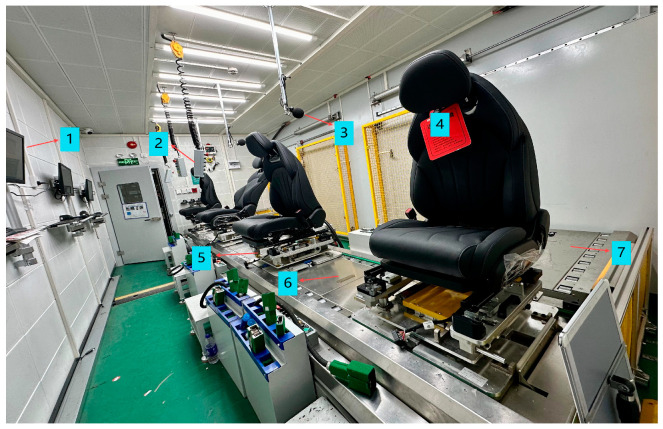
Seat testing line. Labels: 1—Display screen, 2—Button box, 3—Sound level meter, 4—Reworked seat, 5—Workstation, 6—Inspection line, 7—Interface line.

**Table 1 sensors-24-07085-t001:** Seat conditions.

Number	Condition	Symbol
1	Horizontal motor back/forward	SFA F to B/SFA R to F
2	Backrest motor back/forward	SBA F to B/SBA F to F
3	Front elevation motor up/down	STA D to U/STA U to D
4	Rear elevation motor up/down	STA U to U/SHA U to D
5	Thigh support motor back/forward	STA R to B/STA R to F
6	Seat cushion/backrest ventilation	SCV/SBV
7	Lumbar support massage	SLM

**Table 2 sensors-24-07085-t002:** System set threshold.

Value	Upper Limit	Lower Limit
Average/Maximum DBA	42	0
Average/Maximum Sone	7	0
Average/Maximum Current	4	0.1

## Data Availability

Data is contained within the article.

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
