# Peer review of "Development and Design of an Online Quality Inspection System for Electric Car Seats"

_sensors, 2024, doi:10.3390/s24217085_

Round 1

Reviewer 1 Report

Comments and Suggestions for Authors

The research deals with the development and testing of a system that detects defective electric seats integrated into a seat assembly line. Although the work is not scientifically sound, the results presented are of interest to practitioners.

General comments:

1) It would be interesting to add some statistics related to defective seats in the industry to highlight the importance of such system development.

2) The redundant checking of both current and noise should be justified, as both clearly indicate the defective seats.

3) It is not clear how the system chooses between two NO conditions when performing the "Data Normalcy Check".

4) Figure 22 shows that the seats being tested are close. How does the noise of nearby faulty seats affect the measurements of normal seats?

Specific comments:

- It is not necessary to summarise the results at the end of the introduction.

- Line 38 should be checked for frequency unit.

- The captions of Figures 13 and 18 should be changed.

- Check lines 581-582

Reviewer 2 Report

Comments and Suggestions for Authors

The manuscript presents an online quality inspection system for electric car seats using LabVIEW, but it lacks clarity regarding its novelty compared to existing studies on noise and electrical fault detection in automotive components. The literature review is superficial, failing to include critical comparisons or notable advancements, such as the application of artificial intelligence (AI).

The methodology for processing sound and current signals is inadequately explained, particularly regarding the choice of algorithms and sampling frequency. Details on the calibration of the anechoic chamber and the system's performance in real-world conditions are also missing. While the paper highlights a 34.84% reduction in testing time, it does not address other critical metrics like fault detection accuracy and system robustness. A more comprehensive evaluation is necessary to validate the system's reliability.

The lack of comparative analysis with existing systems undermines the claims of novelty. Additionally, grammatical errors and awkward phrasing affect readability, and the caption for Figure 18 contains placeholder text that needs replacing.

Finally, the conclusion should better summarize the findings' practical implications, and the outlook section requires concrete recommendations for future research. Combining these elements into a cohesive section would strengthen the manuscript.

Comments on the Quality of English Language

no comment

Reviewer 3 Report

Comments and Suggestions for Authors

The paper related to the "Development and Design of an Online Quality Inspection Sys- 2 tem for Electric Car Seats" following suggestions below mentioned:

1- Rewrite the abstract and include the introduction, research question, methodology, solution and results.

2—In the introduction section, citations are not in good style. For example, "Wu, H. et al." is not a good way to cite. It's my suggestion to cite it as per the rule.

3- Revise the figure and table title in the whole paper.

4- Proofread the paper again. Have some typo mistakes.

Comments on the Quality of English Language

Need to be improved.

Reviewer 4 Report

Comments and Suggestions for Authors

Thank you for this impressive work on the development of the online quality inspection system for electric car seats. The integration of LabVIEW for real-time data collection and analysis is commendable, and the improvements in production cycle efficiency are significant.

However, I would like to suggest a couple of enhancements that could add depth to your study:

- Mathematical Model: It would be beneficial to include the mathematical model used to represent the dynamics of the electric car seat system. A well-defined mathematical model can provide insights into the underlying mechanics and performance characteristics of the seat, aiding in better understanding and potential future improvements. This could also facilitate the validation of the system’s outputs against theoretical predictions.

- Comfort Assessment: Additionally, could you explore the feasibility of assessing the comfort level of passengers or drivers before and after the implementation of your inspection system? Understanding how the quality inspection processes influence the overall comfort of the seats could provide valuable feedback for both production and design teams. This assessment could include subjective surveys as well as objective measurements of seating posture or pressure distribution.

Comments on the Quality of English Language

I think the English is sufficient and needs minor check 

Round 2

Reviewer 2 Report

Comments and Suggestions for Authors

No more questions. 

Reviewer 3 Report

Comments and Suggestions for Authors

All comments are cooperated as per suggestion